# Novel In Vitro Models for Cell Differentiation and Drug Transport Studies of the Human Intestine

**DOI:** 10.3390/cells12192371

**Published:** 2023-09-27

**Authors:** Randy Przybylla, Mathias Krohn, Marie-Luise Sellin, Marcus Frank, Stefan Oswald, Michael Linnebacher

**Affiliations:** 1Molecular Oncology and Immunotherapy, Clinic of General Surgery, Rostock University Medical Centre, 18057 Rostock, Germany; randy.przybylla@med.uni-rostock.de (R.P.); mathias.krohn@med.uni-rostock.de (M.K.); 2Research Laboratory for Biomechanics and Implant Technology, Department of Orthopedics, Rostock University Medical Centre, 18057 Rostock, Germany; marie-luise.sellin@med.uni-rostock.de; 3Medical Biology and Electron Microscopy Centre, 18057 Rostock, Germany; marcus.frank@med.uni-rostock.de; 4Department Life, Light and Matter, University of Rostock, 18059 Rostock, Germany; 5Institute of Pharmacology and Toxicology, Rostock University Medical Centre, 18057 Rostock, Germany; stefan.oswald@med.uni-rostock.de

**Keywords:** 2D cell lines, ADME, differentiation, drug development, drug transport, intestinal epithelial cells, in vitro, jejunum, small intestine

## Abstract

The most common in vitro model for absorption, distribution, metabolism, and excretion (ADME) purposes is currently the Caco-2 cell line. However, clear differences in gene and protein expression towards the small intestine and an, at best, fair prediction accuracy of intestinal drug absorption restrict the usefulness of a model for intestinal epithelial cells. To overcome these limitations, we evaluated a panel of low-passaged patient-derived colorectal cancer cell lines of the HROC collection concerning similarities to small intestinal epithelial cells and their potential to predict intestinal drug absorption. After initial screening of a larger panel, ten cell lines with confluent outgrowth and long-lasting barrier-forming potential were further characterized in close detail. Tight junctional complexes and microvilli structures were detected in all lines, anda higher degree of differentiation was observed in 5/10 cell lines. All lines expressed multiple transporter molecules, with the expression levels in three lines being close to those of small intestinal epithelial cells. Compared with the Caco-2 model, three HROC lines demonstrated both higher similarity to jejunal epithelial tissue cells and higher regulatory potential of relevant drug transporters. In summary, these lines would be better-suited human small intestinal epithelium models for basic and translational research, especially for ADME studies.

## 1. Introduction

The most common route for drug administration is the oral route [1]. Within the gastrointestinal system, the duodenum and jejunum play the most crucial roles in drug pharmacokinetics due to their large surface area and the abundance of transporters and metabolizing enzymes [2,3]. Therefore, appropriate 2D cell-based models both mimicking the human intestinal epithelium in vitro and predicting the absorption of novel drug candidates are highly important. In drug discovery, the Caco-2 model has been widely used for ADME (absorption, distribution, metabolism, and excretion) purposes. This cell line is known to develop a well-differentiated polarized cell monolayer and to form tight junctions (TJ) and microvilli structures in culture [4]. However, major drawbacks of the Caco-2 model are its adenocarcinoma origin and its considerably different gene and protein expression patterns of drug transporters (DTs) and nuclear receptors (NRs) compared with those of jejunal tissue [5]. Hence, Caco-2 cells are not fully representative of human small intestinal physiology.

The small intestinal epithelium is crucial for the absorption of nutrients and drugs and protects the gut against harmful xenobiotics by forming a tight barrier. Differentiated cell types mediate these functions: absorptive (enterocytes), enteroendocrine (EECs), mucosecreting (goblet cells, GCs), and Paneth cells (PCs) [6], with enterocytes and GCs being the most prevalent [7,8]. Several uptake (solute carrier transporters; SLCs) and ATP-binding cassette (ABC) efflux transporters are located in the apical membrane of enterocytes. SLCs facilitate drug absorption and are the most abundant transporters in the small intestine [9], while ABC transporters limit intestinal absorption and promote the excretion of endogenous substances [10]. In addition, NRs function as modulators of drug-metabolizing enzymes and DT expression [11].

In a previous study, we described a panel of ten 2D colorectal cancer cell lines with long-lasting barrier integrity due to their basal activity and the induction potential of Cytochrome P450 (CYP) 3A4. Of those lines, two candidates with higher CYP3A4 induction potentials than Caco-2 cells were identified [12]. The aim of the present study was to deepen this analysis by assessing intestinal cell differentiation and DT regulation at the transcriptional and protein levels. Additionally, morphological and functional cell characteristics were visualized by immunofluorescent staining and scanning electron microscopy (SEM) as a measure of cellular differentiability.

## 2. Materials and Methods

### 2.1. Chemicals and Reagents

Rifampicin (RIF) was obtained from Carl Roth (Karlsruhe, Germany), and vitamin D3 (VD3) was obtained from Hycultec (Beutelsbach, Germany). Stock solutions of RIF (121.5 mM) and VD3 (1 mM) were prepared in DMSO and ethanol, respectively. The final concentration of DMSO was 0.1%.

### 2.2. Cell Culture

All ten HROC (Hansestadt Rostock, colorectal cancer) cell lines were obtained from the HROC collection [13]. The presented cell panel consists of eight lines derived from human colon cancer; further, two lines were from human rectal cancer. These lines were used with low passage numbers but are immortal and have been tested before for application in standard in vitro assays and for fulfilling basic requirements for intestinal in vitro models [12].

Caco-2 cells were purchased from CLS (Eppelheim, Germany). Cell culture was performed as previously described [12].

For functional studies, cells were treated with 20 µM RIF and 100 nM VD3, respectively. Controls were treated with 0.1% DMSO or ethanol. Cells were harvested after incubation for 24 h, 48 h, and 72 h.

For gene expression analysis and protein quantification, cells were seeded in 6-well plates (Sarstedt, Nümbrecht, Germany) at a density of 5 × 10^5^ and were grown until reaching 60–70% confluence and ten-day post-confluence, respectively. Cells were scraped and centrifuged at 4 °C at 1200 rpm for 7 min, shock-frozen with liquid nitrogen, and stored at −20 °C for further analysis.

### 2.3. Gene Expression Analysis

Total RNA was isolated using Universal RNA Purification Kit (Roboklon, Berlin, Germany) and following the manufacturer’s protocol. RNA quantity was assessed by using NanoDrop 1000 spectrophotometer and ND-1000 spectrophotometer V3.7.1 software (Thermo Fisher Scientific, Waltham, MA, USA). In total, 1000 ng of RNA was reverse transcribed into cDNA using a reverse transcriptase (200 U/µL, Bioron, Römerberg, Germany). Quantitative real-time PCR (qRT-PCR) was performed using the Sibir Rox Hot Mastermix (Bioron) and 10 µM each of the sense and antisense primers according to the manufacturer’s protocol (Table 1). Samples were run on a ViiA7 Real-Time PCR system (Applied Biosystems, Thermo Fisher Scientific) using the PCR conditions 50 °C, 2 min, and 95 °C, 10 min, followed by 35 cycles of 95 °C, 15 s; 58 °C, 1 min; and 70 °C, 45 s and melting curves of 95 °C, 15 s; 60 °C, 15 s; and 95 °C, 15 s. Ct-values were normalized to GAPDH, and relative mRNA expression levels were calculated using the 2^−∆∆Ct^ method. For culture time-dependent gene expression analysis, three normal epithelium biopsies from human small intestine (obtained from the BioBank Rostock) were pooled and included as reference.

### 2.4. Immunocytochemistry

Thirteen-millimeter-diameter cover slips (Carl Roth) were sterilized with absolute ethanol and placed in 24-well plates (Sarstedt). Cells (10^4^ cells per well) were seeded and grown until reaching confluence. HROC lines differed in growth rates [12]. Despite lower doubling times being observed in a few models, all lines were able to reach confluence in 96-well plates within 96 h after seeding of the appropriate amount of cells per well. All samples were fixed with 4% paraformaldehyde (Carl Roth) for 10 min at RT (room temperature); only for Zonula occludens-1 (ZO-1) was staining fixation performed with 1:1 methanol/acetone for 20 min at −20 °C. After rinsing twice with PBS, samples were permeabilized with 0.5% Triton X-100 (AppliChem, Darmstadt, Germany) for 5 min at RT. Cells were washed and incubated with primary antibodies (diluted in PBS, given in Table 2) for one hour at RT and then overnight at 4 °C, protected from light. Cells were washed, and Hoechst 33,342 (diluted 1:250 in PBS; Hycultec) was incubated for 5 min in the dark at RT. For imaging, the CytoViva^®^ Enhanced Darkfield microscope system (CytoViva, Inc., Auburn, AL, USA) with a 60x oil objective and Ocular Software V2.0 (QImaging Ocular Software, Surrey, British Columbia, Canada) was used. Contrast adjustment and image overlay was performed using Adobe Photoshop CS5 (Adobe Inc., San Jose, CA, USA).

### 2.5. Protein Quantification by Liquid Chromatography–Mass Spectrometry/Mass Spectrometry (LC-MS/MS)

Protein abundance of peptide transporter 1 (PEPT1), organic anion-transporting polypeptide 2B1 (OATP2B1), permeability glycoprotein (P-gp), and breast cancer resistance protein (BCRP) were determined using a validated LC-MS/MS-based targeted proteomics method [14,15,16].

Cells were lysed, and the membrane protein fraction was extracted using the ProteoExtract Native Membrane Protein Extraction kit (Merck KGaA, Darmstadt, Germany) according to the manufacturer’s protocol. The obtained membrane fraction was subjected to determination of the whole protein concentrations using the bicinchoninic acid assay (Thermo Fisher Scientific) and was adjusted to a maximum protein concentration of 2 mg/mL. Subsequently, 100 µL of each membrane fraction were mixed with 10 µL of dithiothreitol (200 mM, Sigma-Aldrich, Taufkirchen, Germany), 40 µL of ammonium bicarbonate buffer (50 mM, pH 7.8, Sigma-Aldrich), and 10 µL of ProteaseMAX™ (1%, *m*/*v*, Promega, Mannheim, Germany) and incubated for 20 min at 60 °C (denaturation). After cooling down, 10 µL of iodoacetamide (400 mM, Sigma-Aldrich) was added and the samples were incubated in a darkened water quench for 15 min at 37 °C (alkylation). For protein digestion, 10 µL of trypsin (trypsin/protein ratio: 1/40, Promega) was added, and the samples were incubated in a water quench for 16 h at 37 °C. Digestion was stopped following the addition of 20 µL of formic acid (10% *v*/*v*, Sigma-Aldrich). All samples were stored at −80 °C until further processing. The samples were centrifuged one more time for 15 min at 16,000× *g* and 4 °C. Finally, 50 µL of the digested membrane fraction was mixed with 50 µL of isotope-labeled internal standard (IS) peptide mix (10 nmol/L of each IS). For preparation of the calibration curves, digested human serum albumin (2 mg/mL) was used as a blank matrix and spiked with reference peptides to reach 0.1–25 nmol/L and with IS (final concentration, 5 nmol/L of each peptides). The following proteospecific peptides were used: P-gp, AGAVAEEVLAAIR; BCRP, SSLLDVLAAR; OATP2B1, SSPAVEQQLLVSGPGK; and PEPT1, TLPVFPK. All peptides (purity > 97–99%) were from Thermo Fisher Scientific.

All sample preparation and digestion steps were performed using Protein LoBind tubes (Eppendorf, Hamburg, Germany). Protein quantification was conducted on a 5500 QTRAP triple quadrupole mass spectrometer (AB Sciex, Darmstadt, Germany) coupled to an Agilent Technologies 1260 Infinity system (Agilent Technologies, Berlin, Germany).

For each peptide, 3–4 mass transitions were monitored; the final protein abundance data (picomoles/mg protein) were calculated by normalization to the total protein content of the isolated membrane fraction.

### 2.6. Scanning Electron Microscopy

SEM was used to examine cell morphology and the extent of villous structures present in confluent HROC and Caco-2 cell monolayers. Cells were seeded on sterilized 13 mm diameter glass cover slips (Plano #7013, Wetzlar, Germany) at a density of 10^4^ cells per well in 24-well plates (Sarstedt). When reaching confluence, cells were washed with PBS containing Ca2+ and Mg2+ on the cover slips and then were fixed with a solution containing 2% glutaraldehyde and 1% paraformaldehyde in 0.1 M sodium phosphate buffer at pH 7.3 and stored at 4 °C. Further processing for SEM started with two washes in 0,1 M PBS followed by an ascending series of ethanol in distilled water (30% for 2 min; 50% for 5 min; and 70%, 90%, and pure ethanol for 15 min each) prior to drying using CO_2_ in an Emitech K850 critical point dryer (Emitech/Quorum Technologies Ltd., East Sussex, UK). Surface conductivity was established by sputter coating the specimens with a gold layer of approximately 10–15 nm thickness in a Leica SCD 500 sputter coater (Leica Microsystems, Wetzlar, Germany). Cell surfaces were examined in a field emission scanning electron microscope (Merlin VP Compact, Carl Zeiss, Oberkochen, Germany) using an acceleration voltage of 5 kV at a working distance of 5–6 mm with a high efficiency secondary electron detector for digital image acquisition (image size of 1024 × 756 pixels).

### 2.7. Data Analysis

For gene expression analysis, QuantStudio™ Real-time PCR software 1.3 (Thermo Fisher Scientific) was used. For statistical analysis between treated and untreated groups and time to confluent growth on alterations of gene expression and protein levels, the paired two tailed Student’s *t*-test was used. Spearman’s rank correlation was computed for gene expression data and protein levels. GraphPad Prism software 9.0.0 (GraphPad Software, Inc., San Diego, CA, USA) was used for calculation.

## 3. Results

### 3.1. Culture Time-Dependent Gene Expression

The effects of the ten-day post-confluence (d10) state on intestine-specific gene expression are shown in Figure 1. Confluence-dependent increases in OCLN, VDR, PXR, OATP2B1, glucose transporter 2 (GLUT-2), UGT1A6, and P-gp mRNAs were most pronounced in HROC60, HROC80 T1 M1, and HROC183 T0 M2. Genes were stably expressed in HROC217 T1 M2, and with the exception of GLUT-2, confluence-dependent changes were also observed for this line. The least effects were observed for HROC60 and HROC383 with expression levels staying distinctly below the small intestinal expression. In HROC43, the VDR, OATP2B1, UGT1A6, and BCRP mRNAs were more highly expressed in the control group than in the post-confluence state. UGT1A6 was barely expressed in HROC126 and absent in HROC383. A high up-regulation of PXR mRNA was observed in HROC183 T0 M2 (73-fold) and HROC80 T1 M1 (52-fold) at d10 post-confluence. OCLN and OATP2B1 were overexpressed in Caco-2 cells in the post-confluence state (25- and 36-fold change). HROC32, HROC126, and Caco-2 lacked sufficient BCRP expression.

A key aspect of novel human intestinal in vitro models’ characterization was the detection of relevant transporter and nuclear receptor expression. A correlation analysis was performed to examine the simultaneous expression of relevant DTs and NRs to deliver evidence for functional gene–gene interaction networks active in these lines. PXR mRNA expression significantly correlated with VDR mRNA (r = 0.709, *p* = 0.018) and GLUT-2 mRNA (r = 0.613, *p* = 0.049, shown in Figure 2). The expression of P-gp mRNA correlated with the mRNAs of PXR (r = 0.662, *p* = 0.03), OATP2B1 (r = 0.646, *p* = 0.035), and UGT1A6 (r = 0.654, *p* = 0.032). Further, a significant correlation was found for VDR and GLUT-2 mRNAs (r = 0.627, *p* = 0.043) at d10 post-confluence. In summary, significant correlations between the expression of selected transporters and NRs were more abundant with an increase in culture time.

### 3.2. Culture Time-Dependent Protein Levels

Protein levels of PEPT1, OATP2B1, P-gp, and BCRP in the basal and d10 post-confluence state are shown in Figure 3. A confluence-dependent increase in PEPT1 protein was observed in all lines and strongly correlated with an increase in culture time (*p* = 0.0004). Referred to human jejunal tissue, higher protein levels of PEPT1 were detected in HROC217 T1 M2 (3.1-fold), HROC60 (2.9-fold), HROC80 T1 M1 (2.9-fold), and HROC32 (2.8-fold) but were lowest in Caco-2 (0.1-fold). PEPT1 protein levels most comparable with human jejunum were found in HROC159 T2 M4 (1-fold), HROC383 (1-fold), HROC126 (0.8-fold), and HROC43 (0.6-fold) in the d10 post-confluence state.

OATP2B1, P-gp, and BCRP protein levels varied between both groups and were not significantly increased with prolonged confluence. Higher OATP2B1 protein levels compared with that in the jejunum were detected in Caco-2 (9-fold), HROC80 T1 M1 (3-fold), HROC32 (1.9-fold), and HROC159 T2 M4 (1.8-fold), while protein levels most similar to those of jejunum were found for HROC43 (1.2-fold), HROC183 T0 M2 (1.4-fold), and HROC239 T0 M1 (0.5-fold) in the d10 post-confluence state.

The P-gp protein had higher abundance in HROC159 T2 M4 (4.2-fold), HROC126 (3.2-fold), HROC183 T0 M2 (1.9-fold), and HROC80 T1 M1 (1.8-fold) compared with the levels in the basal state, with protein levels most similar to those of the jejunum in HROC239 T0 M1 (0.7-fold), HROC217 T1 M2 (0.4-fold), HROC43 (1.8-fold), and HROC80 T1 M1 (1.8-fold). The lowest protein levels were detected in Caco-2 (0.3-fold) and HROC32 (0.2-fold). No expression was found for HROC60 and HROC383 in the d10 post-confluence state.

The protein abundance of BCRP was highest in HROC43 (2.1-fold), HROC183 T0 M2 (0.2-fold), and Caco-2 (0.2-fold) and was absent in the remaining lines at d10 post-confluence. However, the BCRP protein levels were closer to the jejunal protein levels in the basal state in HROC217 T1 M2 (0.5-fold), HROC383 (0.5-fold), and HROC239 T0 M1 (0.2-fold) compared with post-confluence.

In summary, the protein levels of PEPT1, OATP2B1, P-gp, and BCRP were higher in HROC43, HROC183 T0 M2, HROC217 T1 M2, and HROC239 T0 M1 compared with those in Caco-2, with protein levels closest to those of human jejunal tissue in HROC43. In total, HROC43 was the single cell line demonstrating sufficient human jejunal-like protein levels of PEPT1, OATP2B1, P-gp, and BCRP. The CYP3A4 and PXR proteins were absent in all lines.

### 3.3. Regulation of Intestinal Drug Transporter Expression

The effects of RIF and VD3 treatment on transporter expression are shown in Figure 4. These data were combined with CYP3A4 activity levels from a recently published data set [12]. In summary, drug treatment led to a high variability of drug response within all HROC lines. There was a stronger regulation of efflux transporters P-gp and BCRP in six HROC lines compared with that of Caco-2 cells. VD3 treatment led to a higher increase in P-gp mRNA in seven HROC lines compared with in Caco-2 cells (1.4-fold change at 72 h). It was highest in HROC43 (6.5-fold change at 72 h) and in HROC60 (3.8-fold change at 72 h). BCRP mRNA levels were also higher upregulated compared with those of Caco-2 (1.4-fold change at 48 h) in five HROC lines when treated with RIF and were highest in HROC32 (4.2-fold change at 72 h), HROC43 (3.3-fold change at 72 h), and HROC60 (3.2-fold change at 72 h). VDR-mediated induction of BCRP mRNA was higher in HROC239 T0 M1 (3.5-fold change at 24 h), HROC32 (2.3-fold change at 48 h), and HROC60 (2.2-fold change at 72 h) compared with that of Caco-2 (1.7-fold change at 48 h).

RIF and VD3 treatment led to higher regulation of CYP3A4 activity and P-gp mRNA in HROC183 T0 M2 and HROC217 T1 M2 compared with that in Caco-2. HROC60 showed comparable VDR-mediated induction of PXR and UGT1A6 expression and stronger P-gp and OATP2B1 expression than Caco-2. The highest PXR-mediated regulation of CYP3A4 activity as well as the highest OATP2B1, UGT1A6, P-gp, and BCRP expression were found in HROC43. Multiple DTs were downregulated in HROC126 and HROC159 T2 M4. In HROC159 T2 M4, RIF treatment led to downregulation of CYP3A4 activity, as well as UGT1A6, P-gp, and PXR expression, whereas VD3 treatment increased CYP3A4 activity but decreased OATP2B1, UGT1A6, P-gp, BCRP, PXR, and VDR expression.

VDR mRNA was increased in HROC32 (2.4-fold change at 48 h), HROC43 (2.1-fold change at 72 h), HROC60 (2.6-fold change at 72 h), and HROC239 T0 M1 (2.1-fold change at 24 h) with VD3 at levels comparable with those of Caco-2 (3.2-fold change at 72 h). VD3-mediated induction of PXR mRNA was highest in HROC43 (3.9-fold change at 72 h) and was further observed for HROC239 T0 M1 (3.5-fold change at 72 h), Caco-2 (3-fold change at 72 h), HROC217 T1 M2 (2.3-fold change at 48 h), and HROC383 (1.7-fold change at 72 h).

In HROC383 cells, the only significant change was a decrease in VDR mRNA after VD3 treatment (0.6-fold change at 72 h). HROC383 cells appeared to have low sensitivity to RIF and VD3 and were found to be comparable with Caco-2. However, a strong increase in PXR mRNA with RIF was observed in Caco-2 (16.6-fold change at 48 h). Overall, HROC43, HROC60, HROC80 T1 M1, HROC183 T0 M2, and HROC217 T1 M2 showed clearly stronger responses in gene regulation subsequent to RIF and VDR treatment than Caco-2.

### 3.4. Cell Morphology and Differentiation

ZO-1 is essential for proper apical surface assembly and is typically localized on discrete sides of cell–cell contact and within the cytoplasm [17]. ZO-1 was expressed at the apical junctions of all confluent HROC cell monolayers (shown in Figure 5). This corresponds well with stable TEER values reported in our previous study [12]. In HROC43, HROC80 T1 M1, HROC159 T2 M4, HROC183 T0 M2, and HROC217 T1 M2, ZO-1 was well organized and showed a consistent spiderweb-like staining pattern, as found in Caco-2 cells (Figure 5K). A thin thread-like pattern was found in HROC32 (Figure 5A). ZO-1 expression was lower in HROC60, HROC126, HROC239 T0 M1, and HROC383 cells (Figure 5C,E,I,J, respectively). However, higher fluorescent signals in particular areas were found for HROC60.

Further, SEM revealed well-developed cell–cell boundaries evidenced by protruding membrane ridges and alterations in villi decoration at the apical side of confluent HROC monolayers, indicating tight cell apposition. Representative photographs and resolution details are shown for HROC60, HROC159 T2 M4, and Caco-2 cells as a reference in Figure 6.

### 3.5. Detection of Intestinal Brush Border

SEM micrographs demonstrated the presence of microvilli on the apical cell surface of all ten HROC cell lines analyzed after growth to cell monolayers (Figure 7). Microvilli structures and surface decoration differ with respect to density and length of microvilli among the ten examined HROC cell lines (Figure 7A–J) and the Caco-2 cells analyzed for reference comparison (Figure 7K). For example, longer villous structures with higher densities compared with Caco-2 cells were observed in HROC183 T0 M2 and HROC217 T1 M2 cells (Figure 7G,H). Microvilli morphology and decoration in HROC80 T1 M1 cells were similar to those of Caco-2 cells (Figure 7D), whereas an altered density and length of microvilli compared with those of Caco-2 was observed in HROC60 cells (Figure 7C). Variations in the decoration with microvilli structures were also found among individual cells of the same cell line (e.g., Figure 7E), indicating that the extent of microvilli development in addition might be influenced by cell state and maturity. To allow for a direct comparison, the HROC cells are depicted with fully developed microvilli decoration (Figure 7A–J) next to the Caco-2 reference cells for which both low and high cell surface microvilli densities are shown (Figure 7K1,K2). In conclusion, all ten HROC cell lines show microvilli structures, which mirror the phenotype of primary human small intestinal epithelial cell monolayers [18].

Intestinal mucus-producing GCs lubricate luminal contents and are essential in the maintenance of intestinal homeostasis [19]. Mucin 2 (MUC2)-producing cells were clearly identified by the perinuclear staining patterns in HROC32, HROC43, HROC60, HROC80 T1 M1, HROC159 T2 M4, HROC183 T0 M2, HROC217 T1 M2, and HROC383 cells (Appendix A). In HROC43, a strongly increased cytoplasmic secretion-like staining pattern was detected (Appendix A). Strong homogenous fluorescence patterns were also observed for HROC60 and HROC159 T2 M4. HROC60 was characterized by fine fluorescent droplets and structures lateral to nuclei, whereas signals were slightly lower in HROC159 T2 M4 with prominently arranged droplets in cytoplasmatic regions (Appendix A). Signals were lower in HROC32, HROC80 T1 M1, HROC183 T0 M2, and HROC217 T1 M2. However, scattered perinuclear fluorescence was observed in these lines with tail-like structures in HROC32 and HROC217 T1 M2 (Appendix A). Unspecific fluorescent patterns were observed in HROC126, HROC239 T0 M1, and Caco-2, with no single cells detected (Appendix A).

PCs secrete antimicrobial peptides and immunomodulating proteins. They are essential in modulating the microbiome and in maintaining intestinal homeostasis [20]. Immunostaining with the PC marker lysozyme (LYZ) led to high cytoplasmatic fluorescence in HROC43, characterized by fine droplets in nuclear and perinuclear regions (Appendix A). Homogenous perinuclear fluorescent staining patterns were found for HROC60, HROC126, and HROC159 T2 M4, characterized by finely granular cytoplasmatic fluorescence in HROC60 (Appendix A). HROC80 T1 M1 und HROC217 T1 M2 demonstrated perinuclear fluorescence patterns with single-side nuclei fitting structures (Appendix A). Lower basic signals were observed for HROC32, HROC183 T0 M2, und Caco-2 (Appendix A). However, partially specific perinuclear fluorescent patterns became evident in these lines. In contrast, a low fluorescence intensity with coarse-dropped structures in the perinuclear and nuclear regions was found for HROC239 T0 M1 (Appendix A). Fluorescence intensity was also less pronounced in HROC383, characterized by a weak perinuclear staining pattern (Appendix A).

EECs secrete a variety of hormones and regulate digestion, appetite, gut motility, and metabolism [21]. Immunofluorescent staining of the EEC marker chromogranin A (CHGA) revealed finely granular fluorescence in all lines with single cells detected in HROC32, HROC43, HROC60, HROC126, HROC183 T0 M2, HROC217 T1 M2, HROC239 T0 M1, and Caco-2 cells (Appendix A). The strongest signals were observed for HROC43 and HROC60. The fluorescence pattern of HROC43 was characterized by strong granular stainings in regions with pronounced perinuclear and nuclear fluorescence (Appendix A). HROC60 showed a more homogenous and intense perinuclear fluorescence pattern with less fluorescent granular patterns (Appendix A). HROC183 T0 M2 showed perinuclear patterns, and HROC217 T1 M2 occasionally nuclear fluorescence in regions with pronounced granular staining (Appendix A). HROC32, HROC126, and HROC159 T2 M4 were characterized by a homogenous cytoplasmatic basic fluorescence that was partly depicted as a perinuclear fluorescence pattern in HROC32 und HROC126 (Appendix A). In Caco-2, the fluorescence intensity was low, with perinuclear fluorescence in regions with pronounced granular patterns (Appendix A). The signals were also low in HROC80 T1 M1; however, granular fluorescence became evident in a high number of nuclei (Appendix A). HROC239 T0 M1 was characterized by a prominent homogenous cytoplasmatic basic fluorescence, partially shown as arrowhead-like structures lateral to nuclei (Appendix A). The lowest signals with a weak pronounced granular pattern and perinuclear fluorescence were observed for HROC383 (Appendix A).

In summary, cell differentiation to specific cell types was strongly pronounced in HROC32, HROC43, HROC60, HROC80 T1 M1, HROC159 T2 M4, HROC183 T0 M2, and HROC217 T1 M2 cells. In contrast, cell differentiation was restricted in HROC126, HROC239 T0 M1, HROC383, and Caco-2 cells.

## 4. Discussion

Intestinal drug transport is mediated by the most prominent transporters PEPT1, OATP2B1, BCRP, and P-gp [22]. In addition, PXR regulates the functions of DTs and DMEs such as UGT1A6 [23]. Due to high CYP3A4 induction by VD3, as previously assessed for a subset of HROC lines, VDR expression and mRNA regulation by VD3 was further examined in the present study. Since the intestinal epithelium is mainly involved in the uptake of key nutrients such as sugars and amino acids, we also assessed the expression of GLUT-2, which is mainly involved in intestinal glucose absorption and thought to mediate efflux of glucose from enterocytes into the blood [24].

Differentiation in secretory IECs (GCs, PCs, and EECs) is of functional relevance in maintaining the digestive and barrier functions of the epithelium [25]. Next to an expression analysis, investigating the differentiation grade of the HROC cell lines is therefore essential for interpreting ADME measurements, especially in the context of drug development. With regard to the lack of mucus producing cells in Caco-2 compared with a normal intestinal epithelium, the detection of goblet cells has special significance for pharmaceutical drug screening. The intestinal mucus layer acts as a barrier and affects transport of lipophilic molecules, thereby strongly influencing drug uptake in vivo [25]. The aim of this study was to identify novel human intestinal in vitro models for studying cell differentiation and drug transport. Therefore, ten cell lines were first identified out of the broad HROC collection and were subsequently characterized concerning their potential to differentiate into intestinal epithelial cells (IECs) and to express intestinal proteins and DTs with and without the addition of regulating drugs—all in comparison with Caco-2 as the classical model. Our data show functional tight junction complexes and defined membrane border ridges as well as the formation of microvilli structures on the apical surface of all cell lines analyzed. This is indicative of enterocytic cell differentiation, intact cell–cell contact, and epithelial barrier integrity. Judged from the nuclear as well as cytoplasmic ZO-1 staining patterns within the lines, TJ complexes and thus intestinal barrier function were omnipresent. Since low ZO-1 expression is reported in poorly differentiated cell lines [26], and the weak signals in HROC126 and HROC383 suggest a lower grade of differentiation of these lines, which might also explain the limited number of relevant IECs detected in the HROC126 and HROC383 monolayers.

Immunostaining for MUC2 revealed cell line-specific differences in fluorescence patterns. Mucin-stream-like patterns in HROC43 differed considerably from that of the other lines. When taking this observation plus the gene expression and protein quantification data into account, this specific line might indeed model the small intestine well, especially jejunal epithelial cells, whereas in particular for HROC60 and HROC159 T2 M4, a prominent intracellular fluorescence pattern was observed. In another human colonic cell line grown to near confluence, MUC2 was both present within the cell layer and was observed to be secreted into the cell medium [27]. MUC2 expression has been described to also vary from colorectal carcinomas of different histological types [28]. Low MUC2 expression may be further explained by poorly differentiated colorectal and rectal adenocarcinoma or advanced tumor stage in CRC [29]. In the small intestine, PCs are located in the crypt next to stem cells and secrete antimicrobial substances such as LYZ [30]. Distinctive LYZ staining patterns were observed for HROC43, HROC60, HROC159 T2 M4, and HROC217 T1 M2. These observations in the colonic HROC cell lines may be explained by the existence of PCs reported for inflammatory bowel diseases, adenoma, and carcinoma [31]. It has further been shown that PCs acquiring stem-like features might trigger intestinal cancer development [32]. For the colon, no PCs but cells analogous to small-intestinal PCs, also with secretory and niche-like functional roles, have been described [33].

EECs comprise approximately 1% of the intestinal epithelium and secrete hormones and peptides to recruit immune cells [34]. Distinctive staining of CHGA was observed for HROC43, HROC60, HROC159 T2 M4, and HROC239 T0 M1. The higher CHGA expression may be explained by the existence of CHGA-positive EEC subtypes in these HROC cell lines [35]. Further, the abundance of intracellular CHGA in HROC159 T2 M4 and HROC239 T0 M1 may be due to the role of CHGA in intracellular calcium homeostasis. However, it should be mentioned that CHGA protein measurements are still challenging due to the known low specificity and cross-reactivities for CHGA and its derived peptides by antibodies [36].

We identified three lines, HROC43, HROC183 T0 M2, and HROC217 T1 M2, which closely resembled in vivo jejunal properties and would thus be very promising intestinal epithelium in vitro models. As monolayer models, these lines differentiated into all relevant IEC types and expressed a variety of DTs, thereby closely resembling normal human intestinal epithelium [6]. Notably, compared with Caco-2, a higher regulation of CYP3A4 activity and P-gp mRNA by RIF and VD3 was also observed. Especially BCRP was highly expressed in these three HROC lines, adding to the range of intestine-specific DTs analyzable in these novel in vitro models.

Another interesting finding of our study was the identification of small intestinal-like gene expression in the rectal cancer cell line HROC239 T0 M1. There, stable OATP2B1 mRNA and protein expression was also observed, which is a little unexpected since OATP2B1 was not detectable in human rectal tissue [37]. Still, HROC239 T0 M1 might be a useful in vitro model for studying rectal drug administration. Despite their colonic origin, HROC lines might differ in mirroring jejunal properties due to different primary tumor localization sides. This might explain jejunal properties remaining in a subset of cell lines and eventually resulting in a higher variance in small intestinal properties among these lines. Our study outlined several drawbacks of the Caco-2 model. First, GC properties were lacking in the Caco-2 model (e.g., low mucin), confirming the findings of others [38]. A lack of GCs can prompt an overestimation of drug permeability (Lea 2015). In contrast, HROC43 showed a distinct mucin expression, indicating GC-like properties of this cell line. Second, much higher OATP2B1 mRNA and protein levels were detected in long-term Caco-2 cultures compared with the intestinal epithelium, again consistent with other studies [5,39]. Third, as recently reported, Caco-2 cells showed low PXR-and VDR-mediated regulation of multiple DTs and CYP3A4 activity [12]. In our previous study, HROC cell lines were incubated for 21 days to perform TEER measurements and FITC dextran permeability tests according to the protocols used for Caco-2 [40,41]. Our current study aimed to standardize a protocol for the HROC lines with shorter culture time periods necessary for differentiation into IECs and application as intestinal epithelial models for transport studies. Thus, these lines were grown for ten days post-confluence instead of the 21 days necessary for Caco-2 [42]. To date, a large number of cell lines derived from human colon carcinomas were successfully established. They vary widely concerning differentiation states, proliferation, and metabolic characteristics. However, most of them do not differentiate under standard culture conditions [43].

In the present study, we clearly expanded the list of well-differentiating lines by identifying three HROC lines able to differentiate into IEC subtypes even under standard culture conditions; the latter might come handy for many researchers in the field. These lines are immortal but low-passaged, limiting culture-induced genetic alterations and sub-clonal varieties, as described for Caco-2 [44]. Further, Caco-2 cells require about three weeks to complete cell differentiation [45], limiting their usefulness for high throughput screening of new drug candidates [46]. To overcome the latter issue, efforts to shorten this time period have repeatedly been undertaken [47,48]. HROC43 and HROC217 T1 M2 exhibited intestinal marker expression within a clearly shorter culture period and may thus be perfectly suited for high-throughput culture protocols.

Our study has several limitations. First, we focused on drug transport and regulation mechanisms. Therefore, applications of the models included for nutrient absorption or further functional studies were not tested. Second, the number of transporters quantified on the protein level is smaller than the number of genes included in the expression analyses. Because measuring protein contents with LC-MS/MS is technically challenging and prone to variations, we focused on the detection of PEPT1, OATP2B1, P-gp, and BCRP. Third, the influence of RIF and VD3 on the protein levels of transporters was not analyzed. However, the findings of our study are new and are adequate to characterize the similarity of several HROC lines to human jejunum cell features. Thus, our data provide potentially useful information for new approaches and further investigations of transporters in selected lines. Further, the interpretation of microvilli formation in HROC lines is limited to the fact that microvilli structures and density may strongly be affected by factors like culture conditions or fluid environment [49]. In addition, SEM analysis further lacks proper quantitative data. However, the cell lines demonstrated little variations in villi length of about 1 µm, which are comparable to the 1 to 3 µm villi lengths of the small intestinal enterocytes’ apical side [50]. At least, SEM images of HROC183 T0 M2 and HROC217 T1 M2 show villi structures superior to those found in Caco-2 when compared with length and density.

The initial focus of our study was the investigation of enterocyte differentiation in the preselected HROC cell lines. The objective criteria for such differentiation are expression of certain genes and microvilli formation. Since OATP2B1 is present in enterocytes and PXR expression increases in differentiated IECS, the quantitative data may serve as markers indicating enterocyte differentiation [51,52]. Except for in HROC60 and HROC383, the OATP2B1 protein was stably expressed, while PXR was absent in HROC383. The high expression of OATP2B1 mRNA and proteins in Caco-2 may verify the high abundance of enterocytes where this cell type is most prevalent [25]. In contrast, lower protein levels of OATP2B1 in HROC60 and, in addition, the low PXR gene expression found for HROC383 may confirm the lower grade of enterocyte differentiation, as shown by SEM analysis. In summary, we characterized a larger panel of novel human intestinal 2D cell models which differ in somatic mutational profiles [12], drug sensitivity, degree of cell differentiation as measured by cell type marker expression, as well as individual regulation pattern of drug metabolizing enzymes and DTs. They thus represent a novel and potentially important tool set for functional-cell-type studies, for ADME purposes and for developing target–drug therapies, with the latter both on academic translational research as well as at a more industrial high-throughput screening level (Table 3).

## 5. Conclusions

In conclusion, the results of our study evidenced that compared with Caco-2, the three human CRC cell lines HROC43, HROC183 T0 M2, and HROC217 T1 M2 displayed higher resemblance to jejunal epithelial tissue and higher regulatory potentials of relevant drug transporters. Therefore, these cell lines may be excellent tools for studying ADME, nutrient absorption, and cell differentiation in the intestinal epithelium.

## Figures and Tables

**Figure 1 cells-12-02371-f001:**
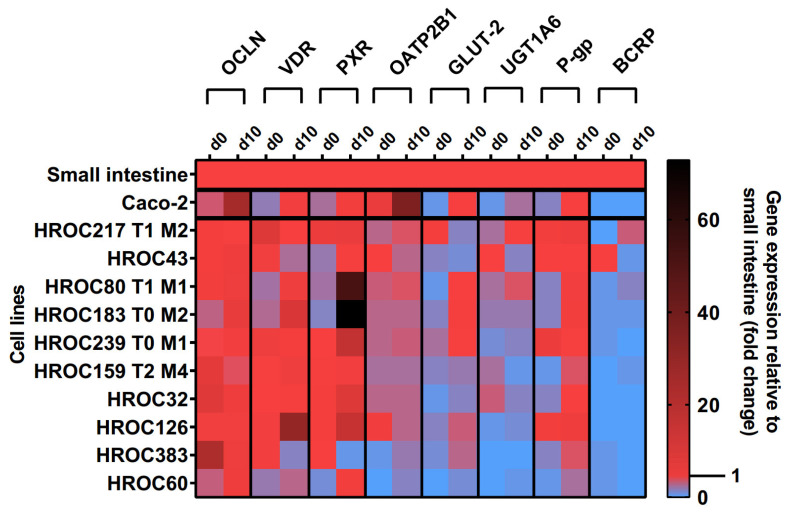
Time-dependent gene expression of selected genes in HROC and Caco-2 cells. Cells were cultured on 6-well plates and then grown until reaching confluence. Fold changes of GAPDH-normalized measurements between control (d0) and ten-day post-confluence growth (d10) were calculated by performing the ∆∆Ct method. Reference tissue was set at 1.0.

**Figure 2 cells-12-02371-f002:**
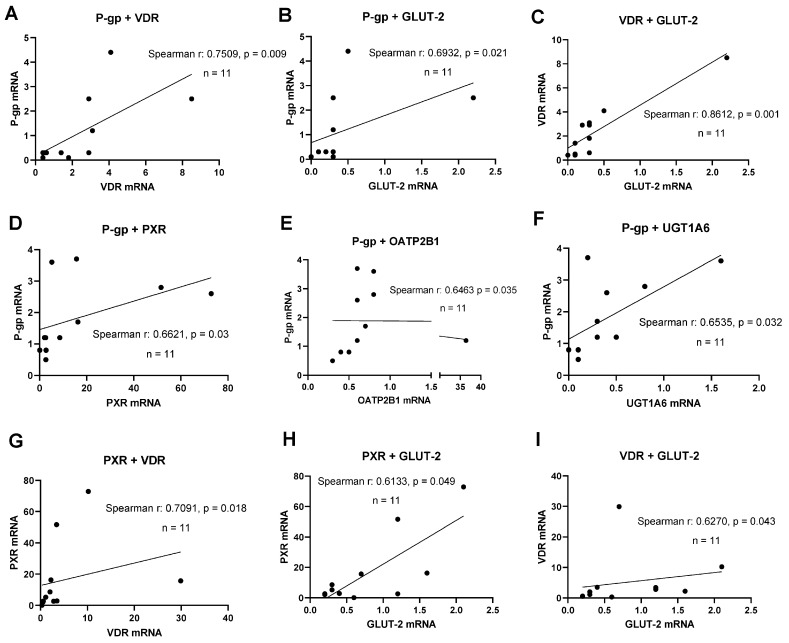
Gene expression correlation analyses. Graphs showing Spearman rank correlation analyses of transporter and nuclear receptor expression levels in HROC and Caco-2 cells in control state (**A**–**C**) and ten-day post-confluence growth (**D**–**I**). Spearman rank correlation coefficient (r) and *p*-values were computed for each analysis. Data are shown as mean from three independent datasets.

**Figure 3 cells-12-02371-f003:**
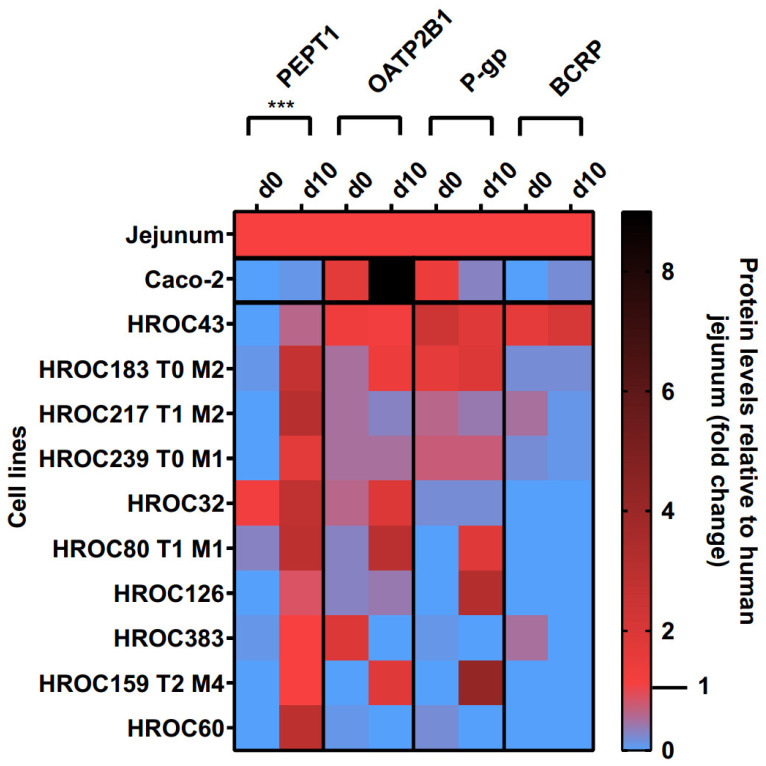
Time-dependent protein expression levels of drug transporters in HROC and Caco-2 cells. Cells were cultured on 6-well plates and then grown until reaching confluence. Protein levels detected by LC-MS/MS were calculated as fold changes relative to human jejunal tissue. Reference tissue was set at 1.0. *** *p* < 0.001 by paired Student’s *t*-test.

**Figure 4 cells-12-02371-f004:**
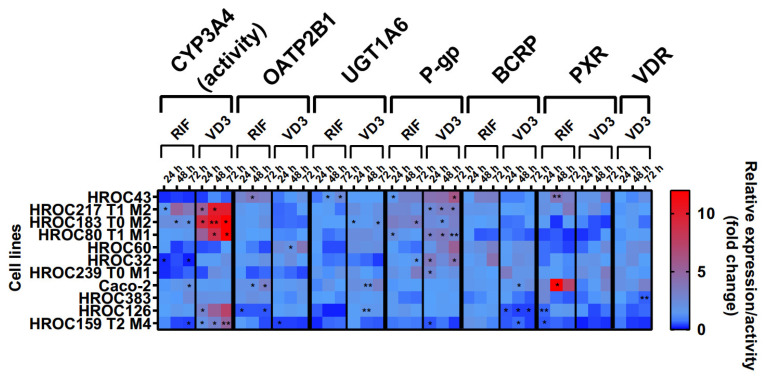
Effects of rifampicin (RIF) and 1a,25-dihydroxyvitamin D3 (VD3) on mRNA expression in HROC and Caco-2 cells. Cells were cultured on 6-well plates and then treated with vehicle (0.1% DMSO or 0.1% ethanol) or inducers (20 µM RIF or 100 nM VD3) for 72 h. Data are presented as mean of three independent experiments. Fold changes in GAPDH-normalized measurements between control and inducer treatment were calculated by performing the 2^−∆∆Ct^ method. Untreated control was set at 1.0. * *p* < 0.01 and ** *p* < 0.005 by paired Student’s *t*-test.

**Figure 5 cells-12-02371-f005:**
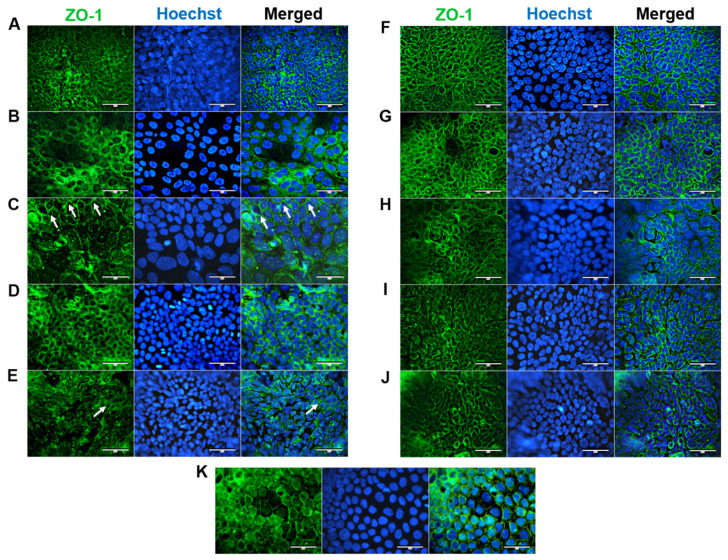
Immunofluorescence analysis of ZO-1 protein localization. (**A**) HROC32, (**B**) HROC43, (**C**) HROC60, (**D**) HROC80 T1 M1, (**E**) HROC126, (**F**) HROC159 T2 M4, (**G**) HROC183 T0 M2, (**H**) HROC217 T1 M2, (**I**) HROC239 T0 M1, (**J**) HROC383, and (**K**) Caco-2. Cells were stained for localization of ZO-1 by direct immunofluorescence. Arrowheads illustrate representative areas of significant fluorescent staining patterns indicative of functional tight junction complexes. Bars = 50 µm.

**Figure 6 cells-12-02371-f006:**
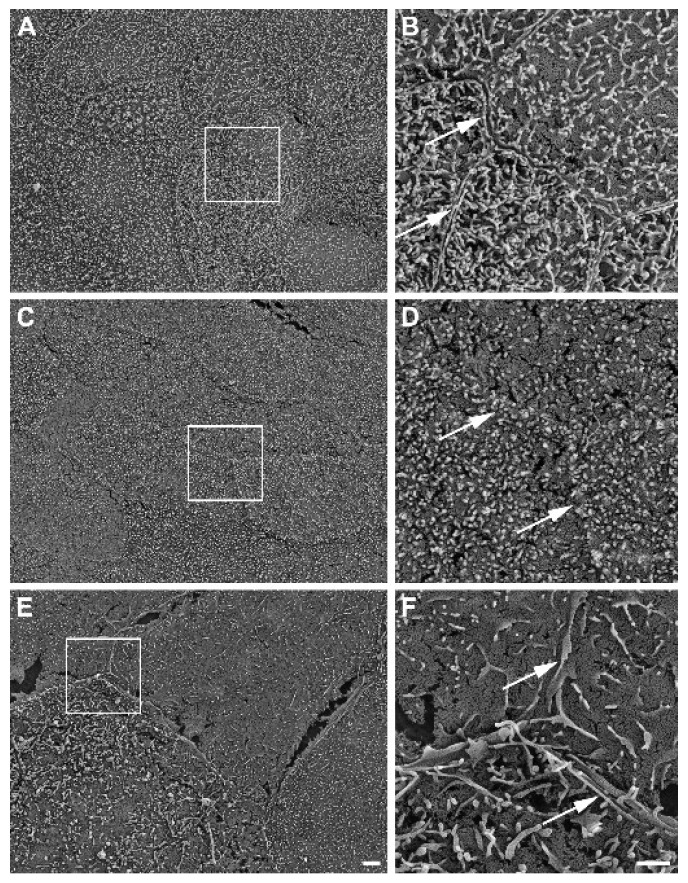
Representative SEM photographs of stratified HROC60 (**A**,**B**) and HROC159 T2 M4 (**C**,**D**) epithelial monolayers in comparison with Caco-2 cells (**E**,**F**). High magnification images from the insert regions demonstrate the close apposition and sealing of membranes at the cell–cell boundaries (arrows). Scale bars are 2 µm for overview images and 1 µm for inserts.

**Figure 7 cells-12-02371-f007:**
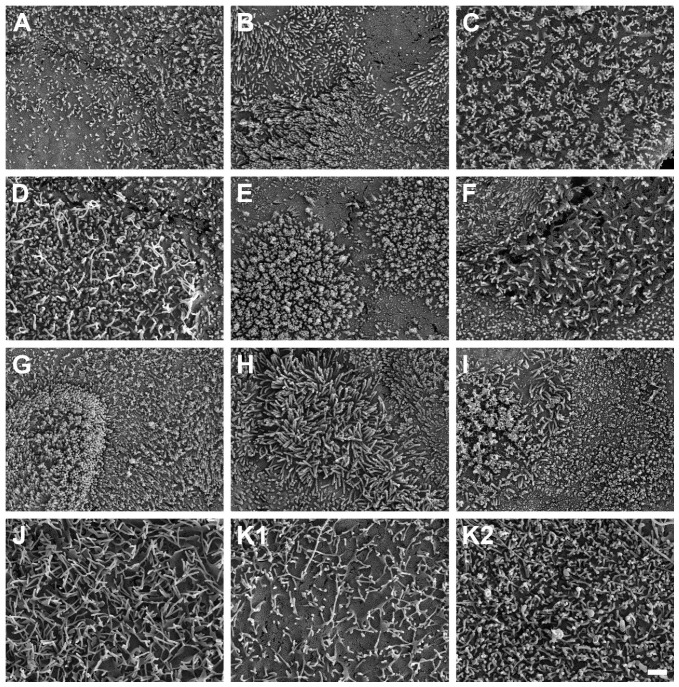
Scanning electron microscope photographs of microvilli formations in HROC and Caco-2 cells after reaching confluence. Cells with a dense coverage of microvilli are depicted: (**A**) HROC32, (**B**) HROC43, (**C**) HROC60, (**D**) HROC80 T1 M1, (**E**) HROC126, (**F**) HROC159 T2 M4, (**G**) HROC183 T0 M2, (**H**) HROC217 T1 M2, (**I**) HROC239 T0 M1, and (**J**) HROC383. Note that microvilli decoration may vary among neighboring cells, and thus for the Caco-2 reference, low (**K1**) and high decorations (**K2**) with microvilli are depicted for reference (**K1**,**K2**). Scale bar is 1 µm.

**Table 1 cells-12-02371-t001:** Primers used for RT-PCR.

Gene Name	5′-3′	Sequences	Product Size (bp)
ABCB1 (P-gp)	Forward Reverse	GCTGTCAAGGAAGCCAATGCCT TGCAATGGCGATCCTCTGCTTC	120
ABCG2 (BCRP)	Forward Reverse	GTTCTCAGCAGCTCTTCGGCTT TCCTCCAGACACACCACGGATA	145
GAPDH	Forward Reverse	GAAGGTGAAGGTCGGAGTC GAAGATGGTGATGGGATTTC	226
SLC2A2 (GLUT-2)	Forward Reverse	ATGTCAGTGGGACTTGTGCTGC AACTCAGCCACCATGAACCAGG	131
SLCO2B1 (OATP2B1)	Forward Reverse	TGGGCACAGAAAACACACCT CGGCTGCCAAAATAGCTCAC	265
OCLN	Forward Reverse	ATGGCAAAGTGAATGACAAGCGG CTGTAACGAGGCTGCCTGAAGT	124
NR1I2 (PXR)	Forward Reverse	GCTGTCCTACTGCTTGGAAGAC CTGCATCAGCACATACTCCTCC	124
UGT1A6	Forward Reverse	GCAAAGCGCATGGAGACTAAGG GGTCCTTGTGAAGGCTGGAGAG	148
NR1I1 (VDR)	Forward Reverse	CGCATCATTGCCATACTGCTGG CCACCATCATTCACACGAACTGG	101

**Table 2 cells-12-02371-t002:** Used antibodies.

Antibody	Host	Company	Dilution	Catalogue Number
anti chromogranin A	mouse	Santa Cruz	1:50	sc-393941
anti lysozyme C	mouse	Santa Cruz	1:50	sc-518012
anti mucin 2	mouse	Santa Cruz	1:50	sc-515032
anti ZO-1	rat	Santa Cruz	1:50	sc-33725

**Table 3 cells-12-02371-t003:** Summary of HROC cell models. Cell line-specific advantages, limitations, and applications for ADME purposes.

Cell Line	Differentiation Grade	Advantages	Limitations	Applications
HROC32	Presence of GCs, PCs, and EECs; villi formation	Protein abundance of PEPT1, OATP2B1, and P-gp; high basal CYP3A4 activity; high efflux effects	Lack of BCRP protein and mRNA, lower levels of GLUT-2 mRNA compared with small intestine, colonic origin	VDR-mediated first-pass metabolism, cell differentiation, cell interaction, HTS
HROC43	Well-developed TJs; closely resembles in vivo jejunal properties; presence of GCs, PCs, and EECs; villi formation	High basal CYP3A4 activity; protein abundance of PEPT1, OATP2B1, P-gp, and BCRP; PXR-/VDR-mediated regulation of CYP3A4 activity + DT mRNA	Lower levels of GLUT-2 mRNA compared with small intestine, colonic origin	First-pass metabolism, permeability, host–microbe interaction, tissue regeneration, HTS
HROC60	Presence of GCs and EECs, villi formation, LYZ expressed	Well-developed TJs, protein abundance of PEPT1, high TEER, stable long-term culture, high VDR-mediated regulation of DTs	Limited degree of differentiation; poor abundance of CYP3A4 and DTs, limited reflection of the in vivo situation; colonic origin	Gut-on-a-chip, tissue regeneration, HTS
HROC80 T1 M1	Presence of GCs, villi formation	Well-developed TJs, PXR-/VDR-mediated regulation of CYP3A4 + DTs	Lack of BCRP protein, limited degree of differentiation (lack of EECs and LYZ), poor PXR-mediated drug response, colonic origin	First-pass metabolism, HTS
HROC126	EECs present, villi formation	VDR-mediated induction of CYP3A4 activity	Limited degree of differentiation (lack of GCs and LYZ), rectal cancer origin, poor PXR-mediated regulation of DTs, VDR overexpressed	VDR-mediated first-pass metabolism, rectal drug administration, HTS
HROC159 T2 M4	GCs present, LYZ expressed, villi formation	Well-developed TJs, high basal CYP3A4 activity, appropriate VDR-mediated regulation of CYP3A4 activity	Poor gene expression and protein levels of multiple DTs, colonic origin	First-pass metabolism, tissue regeneration, HTS
HROC183 T0 M2	Closely resembles in vivo jejunal properties, presence of GCs and EECs, LYZ expressed, villi formation	Well-developed TJs, VDR-mediated induction of CYP3A4 activity, appropriate PXR-/VDR-mediated regulation of CYP3A4 activity and P-gp mRNA	Low basal CYP3A4 activity, colonic origin	PXR-mediated first-pass metabolism, HTS
HROC217 T1 M2	Closely resembles in vivo jejunal properties, presence of GCs and EECs, LYZ expressed, villi formation	Well-developed TJs; protein abundance of PEPT1, OATP2B1, P-gp, and BCRP; appropriate PXR-/VDR-mediated regulation of CYP3A4 activity and P-gp mRNA	Lower protein levels of OATP2B1, P-gp, and BCRP compared with jejunum; colonic origin	PXR-/VDR-mediated first-pass metabolism, HTS
HROC239 T0 M1	Closely resembles in vivo jejunal properties, presence of EECs, villi formation	Protein abundance of PEPT1, OATP2B1, P-gp, and BCRP	Rectal cancer origin; PXR overexpressed; low UGT1A6 mRNA expression; lack of GCs and PCs; colonic origin; lower protein levels of OATP2B1, P-gp and BCRP compared with jejunum	Rectal drug administration, HTS
HROC383	Presence of GCs and villi formation	Protein abundance of PEPT1 and OATP2B1	Limited degree of differentiation, poor PXR- and VDR-mediated drug response, poor abundance of OATP2B1, UGT1A6 not expressed, colonic origin	HTS

## Data Availability

Data is contained within the article or Appendix A of this article.

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
