# Peer review of "Novel In Vitro Models for Cell Differentiation and Drug Transport Studies of the Human Intestine"

_cells, 2023, doi:10.3390/cells12192371_

Round 1

Reviewer 1 Report

General comments:

1)     The authors selected ten HROC cell lines. But there is no description about HROC in this manuscript, only citing a related paper (Mullins et al. 2019). Adding brief introduction of this cell line - for example, origins and general characteristics- is preferable.

2)     This reviewer feels that the interpretation of the experimental results by the authors is not always in good agreement with the data shown by figures and photographs. Authors should explain more precisely how they pulled the conclusion from the data. Important parts in the figures or photos to be carefully observed, key points necessary to draw a conclusion, should be emphasized using arrowheads or other markers. Most of the arrowheads already placed in the photos (Figs. 1, 3-5) are not very useful, because there is no guidance of the meaning of the arrowheads. Accurate use of arrowhead and explanation of the arrowheads are necessary.

3)     In this manuscript the HROC cell lines were characterized from two points of view. The first one is “presence and diversity of differentiated cells”, and the other one is “expression of functional molecules”.  However, it seems that these two issues are not well linked or engaged. Analyzing “expression of drug transporters, metabolic enzymes, and related transcriptional factors” makes sense because they are important players in drug absorption and metabolism. But what is the meaning of investigating “differentiation rate of cell lines” in constructing a cell culture model system for drug absorption and metabolism?  Please consider rearrangement of the structure of the manuscript to clarify the purpose of this study or, at least, add some sentences to explain this issue.

4)     The final Table 3 is informative and will be useful for researchers. However, Table 3 includes information of two different aspects together; one is about “differentiated cells” and the other is about “functional molecules”. Separating the two parts will be better to compare the characteristics of each cell line. I would appreciate if you could consider this suggestion.

Major comments:

1.     Lines 169- (Fig.1) : It is not possible to evaluate integrity of the cell monolayers only by observing the mesh structure of immunostaining of ZO-1 in the photos. The authors should examine the leakiness of monolayers by TER (transepithelial electric resistance) measurement or penetration experiment using small molecular compounds such as Lucifer Yellow. If you think of evaluating drug transport across the epithelial cell monolayers, this is important.

2.     Lines 196- (Fig.3): Stained pattern of MUC2 in HROC43 cells (something like a stream of mucin?) seems to be different from that of other cells.  In HROC60 and 159 cells, for example, MUC 2 is likely to be stained intracellularly. Some explanation is needed.

3.     Line 217- (Fig.4): Lysozyme was distinctively stained in HROC60, 159, and 217 cells. As already known Paneth cells are located in the crypt region next to stem cells, and the cell ratio is small in the intestinal epithelium. In the three cells mentioned above PC seems to be abundant in the cell monolayers. Some explanation will be necessary. By the way, did you detect stem cells? Personally, I am curious about the possible existence of stem cells in HROC cell lines.

4.      Lines238-(Fig. 5):Photos of 159 and 239 suggest that many cells have intracellular chromogranin. The rate of enteroendocrine cells in the intestinal epithelium is usually less than a few percent. I am wondering why EEC is so abundant in these cell lines. Some explanation is necessary.

5.      Lines 264: The authors concluded from immunofluorescence analysis that cell differentiation was strongly pronounced in the seven cell lines such as HROC32, while that was restricted in the three cell lines such as HROC126. The authors should show the criteria or objective reasons to divide the cell lines to two groups, because there are no quantitative data with Fig.1-5.

6.     Lines 269- (Fig.6): As the authors mentioned (lines 279-), cell state and maturity influence the extent of microvilli development. Since formation and morphology of microvilli on the cultured epithelial cells are so changeable and diverse depending on culture conditions, it must be difficult to say something from a small number of photos. The validity of comparing microvilli structure of the eleven cell lines based on Fig.6 should be more clearly explained.

7.     Lines 293-(Fig. 7): Why did you choose these eight molecules for gene expression experiments? Choosing drug transporters, drug metabolizing enzymes, and related transcriptional factors is reasonable to search for cell lines usable to construct a model system for drug transport, but why VDR and GLUT2?  Please give reasons.

8.     Lines 312-(Fig. 8):How would you evaluate “the correlations between transporters and nuclear receptors” in relation to the “characterization of cell lines”? Please add some sentences to explain the meaning of this correlation analysis.

9.     Lines 325- (Fig. 9): PEPT1 newly appears here. Was it simply used as a good example of drug transporters showing very clear time-dependent expression? Please give some supplementary explanation.

You might think you had already answered to some of my questions or comments in the discussion section, but I would appreciate if you could carefully read the above questions/comments and make appropriate revision.

Minor comments

1.     Lines 73 and 95: “104” cells should be corrected (to the 4th power), “105”(to the 5th power)

2.     Lines 95-96, “and grown until reaching confluence”: How long does it take to reach confluence? Is there any difference in growing rate among the cell lines?

3.     Some of the molecule’s names are shown only in abbreviated designation. Full names of such molecules should be added to “Abbreviations” at the end of the text.

4.     Table 3; Applications for ADME purposes in the column:  “drug and nutrient absorption” is entered in all of the ten cell lines. To compare the ten cell lines this is meaningless, rather ruins the simplicity of the Table. How about deleting this phrase and adding a statement about “the usefulness of all the ten cell lines in studying drug and nutrient absorption” in the text (to the last paragraph - final conclusion in lines 466-)?

Author Response

Answers to Reviewer 1

1)     The authors selected ten HROC cell lines. But there is no description about HROC in this manuscript, only citing a related paper (Mullins et al. 2019). Adding brief introduction of this cell line - for example, origins and general characteristics- is preferable.

Thank you for your feedback. A brief introduction concerning origins and general characteristics of the HROC lines was added to the manuscript:

“The presented cell panel consists of eight lines derived from human colon cancer, further two lines were from human rectal cancer. These lines were used in low passage numbers but are immortal and have before been tested to be applicable for standard in vitro assays and for fulfilling basic requirements for intestinal in vitro models (https://doi.org/10.3390/ijms23179861).“

2)     This reviewer feels that the interpretation of the experimental results by the authors is not always in good agreement with the data shown by figures and photographs. Authors should explain more precisely how they pulled the conclusion from the data. Important parts in the figures or photos to be carefully observed, key points necessary to draw a conclusion, should be emphasized using arrowheads or other markers. Most of the arrowheads already placed in the photos (Figs. 1, 3-5) are not very useful, because there is no guidance of the meaning of the arrowheads. Accurate use of arrowhead and explanation of the arrowheads are necessary.

We agree with your comment and added explanation of the arrowheads placed in the Figures 5, S1-3 as follows:

Figure 5: Arrowheads illustrate representative areas of significant fluorescent staining patterns indicative of functional tight junction complexes.

Figure S1: Arrowheads illustrate representative areas of significant fluorescent staining patterns indicative of GCs.

Figure S2: Arrowheads illustrate representative areas of significant fluorescent staining patterns indicative of PCs.

Figure S3: Arrowheads illustrate representative areas of significant fluorescent staining patterns indicative of EECs.

3)     In this manuscript the HROC cell lines were characterized from two points of view. The first one is “presence and diversity of differentiated cells”, and the other one is “expression of functional molecules”.  However, it seems that these two issues are not well linked or engaged. Analyzing “expression of drug transporters, metabolic enzymes, and related transcriptional factors” makes sense because they are important players in drug absorption and metabolism. But what is the meaning of investigating “differentiation rate of cell lines” in constructing a cell culture model system for drug absorption and metabolism?  Please consider rearrangement of the structure of the manuscript to clarify the purpose of this study or, at least, add some sentences to explain this issue.

Thank you for this helpful advice. We decided to rearrange the structure of the manuscript accordingly. Sections concerning cell differentiation were moved to the end of the manuscript and the immunofluorescence images of MUC2, LYZ and CHGA were moved to the supplementary material part. We think, that this will also increase the readability. We further added an explanation to the discussion part to clarify the link between the expression analysis and differentiation grade of the HROC lines as follows:

“Differentiation in secretory IECs (GCs, PCs and EECs) are of functional relevance in maintaining the digestive and barrier functions of the epithelium (doi: 10.1007/978-3-319-16104-4_13). Next to expression analysis, investigating the differentiation grade of the HROC cell lines is therefore essential for interpreting ADME measurements, especially in the context of drug development. With regard to the lack of mucus producing cells in Caco-2 compared to normal intestinal epithelium, the detection of goblet cells has special significance for pharmaceutical drug screening. The intestinal mucus layer acts as a barrier and affects transport of lipophilic molecules, thereby strongly influencing drug uptake in vivo (ISBN : 978-3-319-15791-7).”  

4)     The final Table 3 is informative and will be useful for researchers. However, Table 3 includes information of two different aspects together; one is about “differentiated cells” and the other is about “functional molecules”. Separating the two parts will be better to compare the characteristics of each cell line. I would appreciate if you could consider this suggestion.

We again agree with this reviewer’s comment and modified Table 3 accordingly. A row containing information about differentiation grade in these lines has been added into Table 3.

Major comments:

  1. Lines 169- (Fig.1) : It is not possible to evaluate integrity of the cell monolayers only by observing the mesh structure of immunostaining of ZO-1 in the photos. The authors should examine the leakiness of monolayers by TER (transepithelial electric resistance) measurement or penetration experiment using small molecular compounds such as Lucifer Yellow. If you think of evaluating drug transport across the epithelial cell monolayers, this is important.

This is an absolutely correct statement and both TEER measurements and examination of leakiness by using FITC-Dextran were performed in a previous study. We cited this publication (Przybylla et al. 2022) in section 3.4. The ZO-1 staining performed in the current study aimed to provide additional evidence for tight junctional complexes in these low FITC Dextran permeable and dense cell monolayers.

  1. Lines 196- (Fig.3): Stained pattern of MUC2 in HROC43 cells (something like a stream of mucin?) seems to be different from that of other cells. In HROC60 and 159 cells, for example, MUC 2 is likely to be stained intracellularly. Some explanation is needed.

The following explanations for different MUC2 staining patterns were added to the discussion part:

“Immunostaining for MUC2 revealed cell line specific differences in fluorescence patterns. Mucin-stream-like patterns in HROC43 differed considerably from that of the other lines. When taking this observation plus the gene expression and protein quantification data into account, this specific line might indeed very good model small intestinal, especially jejunal epithelial cells. Whereas in particular for HROC60 and HROC159 T2 M4, a prominent intracellular fluorescence pattern was observed. In another human colonic cell line grown to near confluence, MUC2 was both present within the cell layer and was observed to be secreted into the cell medium (https://doi.org/10.1042/bj3151055). MUC2 expression has been described to also vary from colorectal carcinomas of different histological types (https://doi.org/10.1002/ijc.2910590302). Low MUC2 expression may be further explained by poorly differentiated colorectal and rectal adenocarcinoma or advanced tumor stage in CRC (https://doi.org/10.3892/ol.2017.6218).”

  1. Line 217- (Fig.4): Lysozyme was distinctively stained in HROC60, 159, and 217 cells. As already known Paneth cells are located in the crypt region next to stem cells, and the cell ratio is small in the intestinal epithelium. In the three cells mentioned above PC seems to be abundant in the cell monolayers. Some explanation will be necessary. By the way, did you detect stem cells? Personally, I am curious about the possible existence of stem cells in HROC cell lines.

Unfortunately, little effort has been put into the detection of stem cells in the HROC lines so far. We added the following explanation to the discussion part:

“In the small intestine, PCs are located in the crypt next to stem cells and secrete antimicrobial substances such as LYZ (https://doi.org/10.1007/s00018-002-8412-z). Distinctive LYZ staining patterns were observed for HROC43, HROC60, HROC159 T2 M4 and HROC217 T1 M2. These observations in the colonic HROC cell lines may be explained by the existence of PCs reported for inflammatory bowel diseases, adenoma and carcinoma (https://doi.org/10.1186/s13000-018-0775-z). It has further been shown that PCs acquiring stem-like features might trigger intestinal cancer development (https://doi.org/10.21203%2Frs.3.rs-2458794%2Fv1). For the colon, no PCs but cells analogous to small-intestinal PCs also with secretory and niche-like functional roles have been described (https://doi.org/S0016-5085(12)00171-0, https://doi.org/10.1073/pnas.1607327113).”

  1. Lines238-(Fig. 5):Photos of 159 and 239 suggest that many cells have intracellular chromogranin. The rate of enteroendocrine cells in the intestinal epithelium is usually less than a few percent. I am wondering why EEC is so abundant in these cell lines. Some explanation is necessary.

The following explanation has been added to the discussion part:

“EECs comprise approximately 1 % of the intestinal epithelium and secrete hormones and peptides to recruit immune cells (doi: 10.1177/1756283X08093943, doi: 10.1007/s10735-010-9302-6). Distinctive staining of CHGA was observed for HROC43, HROC60, HROC159 T2 M4 and HROC239 T0 M1. Higher CHGA expression may be explained by the existence of CHGA-positive EEC subtypes in these HROC cell lines (https://doi.org/10.1016/S0092-8674(01)00459-7). Further, abundance of intracellular CHGA in HROC159 T2 M4 and HROC239 T0 M1 may be due to the role of CHGA in intracellular calcium homeostasis. However, it should be mentioned that CHGA protein measurements are still challenging due to a known low specificity and cross-reactivities for CHGA and its derived peptides by antibodies (https://doi.org/10.1016/j.bcp.2018.04.009).”

  1. Lines 264: The authors concluded from immunofluorescence analysis that cell differentiation was strongly pronounced in the seven cell lines such as HROC32, while that was restricted in the three cell lines such as HROC126. The authors should show the criteria or objective reasons to divide the cell lines to two groups, because there are no quantitative data with Fig.1-5.

We added the following explanation to the discussion part. We explained, that initially, the focus was on the investigation of enterocyte differentiation.

“The initial focus of our study was the investigation of enterocyte differentiation in the preselected HROC cell lines. Objective criteria for such differentiation are both expression of certain genes and microvilli formation. Since OATP2B1 is present in enterocytes and PXR expression increases in differentiated IECS, the quantitative data may serve as markers indicating enterocyte differentiation (https://doi.org/10.1038/sj.clpt.6100056, https://doi.org/10.11131/2016/101199). Except for HROC60 and HROC383, OATP2B1 protein was stably expressed, while PXR was absent in HROC383. High expression of OATP2B1 mRNA and protein in Caco-2 may verify the high abundance of enterocytes where this cell type is most prevalent (ISBN : 978-3-319-15791-7). In contrast, lower protein levels of OATP2B1 in HROC60 and in addition, low PXR gene expression found for HROC383 may confirm lower grade of enterocyte differentiation as shown by SEM analysis.”

  1. Lines 269- (Fig.6): As the authors mentioned (lines 279-), cell state and maturity influence the extent of microvilli development. Since formation and morphology of microvilli on the cultured epithelial cells are so changeable and diverse depending on culture conditions, it must be difficult to say something from a small number of photos. The validity of comparing microvilli structure of the eleven cell lines based on Fig.6 should be more clearly explained.

We agree with that comment and added the following concerning study limitations to the discussion part:

“Further, the interpretation of microvilli formation in HROC lines is limited to the fact that microvilli structures and density may strongly be affected by factors like culture conditions or fluid environment (https://doi.org/10.1038/ncomms9871). In addition, SEM analysis further lacks proper quantitative data. However, the cell lines demonstrated little variations in villi length of about 1 µm, which is comparable to the 1 to 3 µm villi  length of the small intestinal enterocytes’ apical side (https://doi.org/10.1083/jcb.201407015). At least, SEM images of HROC183 T0 M2 and HROC217 T1 M2 show villi structures superior to those found in Caco-2 when compared to length and density.”

  1. Lines 293-(Fig. 7): Why did you choose these eight molecules for gene expression experiments? Choosing drug transporters, drug metabolizing enzymes, and related transcriptional factors is reasonable to search for cell lines usable to construct a model system for drug transport, but why VDR and GLUT2? Please give reasons.

According to this advice, the discussion part was supplemented by the following explanations:

“Intestinal drug transport is mediated by the most prominent transporters PEPT1, OATP2B1, BCRP and P-gp (https://doi.org/10.1515/hsz-2016-0259). In addition, PXR regulates functions of DTs and DMEs such as UGT1A6 (ISBN 978-0-12-802447-8). Due to high CYP3A4 induction by VD3 as previously assessed for a subset of HROC lines, VDR expression and mRNA regulation by VD3 was further examined in the present study. Since the intestinal epithelium is mainly involved in the uptake of key nutrients such as sugars and amino acids, we also assessed the expression of glucose transporter 2 (GLUT-2) which is mainly involved in intestinal glucose absorption and thought to mediate efflux of glucose from enterocytes into the blood (https://doi.org/10.1111/j.1432-1033.1994.tb18550.x).”

  1. Lines 312-(Fig. 8):How would you evaluate “the correlations between transporters and nuclear receptors” in relation to the “characterization of cell lines”? Please add some sentences to explain the meaning of this correlation analysis.

Explanations for the meaning of this correlation analysis was added to the result section as follows:

“Key aspect of novel human intestinal in vitro models’ characterization was the detection of relevant transporter and nuclear receptor expression. Correlation analysis was done to examine the simultaneous expression of relevant DTs and NRs to deliver evidence for functional gene-gene interaction networks active in these lines.“

  1. Lines 325- (Fig. 9): PEPT1 newly appears here. Was it simply used as a good example of drug transporters showing very clear time-dependent expression? Please give some supplementary explanation.

Thank you for this comment. PEPT1 was simply used, because it is one of the most important transporters involved in drug absorption. Explanation for choosing PEPT1 as a relevant intestinal drug transporter was now added to the discussion together with the brief explanation of choosing the remaining transporters and enzymes.

You might think you had already answered to some of my questions or comments in the discussion section, but I would appreciate if you could carefully read the above questions/comments and make appropriate revision.

Minor comments

  1. Lines 73 and 95: “104” cells should be corrected (to the 4th power), “105”(to the 5th power)

These changes were made in order to your advice. We apologize for this error.

  1. Lines 95-96, “and grown until reaching confluence”: How long does it take to reach confluence? Is there any difference in growing rate among the cell lines?

The manuscript was supplemented as follows:

“HROC lines differ in growth rates (https://doi.org/10.3390/ijms23179861). Despite lower doubling times were observed in a few models, all lines were able to reach confluence in 96-well plates within 96 h after seeding of the appropriate amount of cells per well.”

  1. Some of the molecule’s names are shown only in abbreviated designation. Full names of such molecules should be added to “Abbreviations” at the end of the text.

The following molecule’s names were added to “Abbreviations” according to this reviewer’s suggestion:

BCRP, CHGA, GLUT-2, LYZ, MUC2, OATP2B1, PEPT1, P-gp

  1. Table 3; Applications for ADME purposes in the column: “drug and nutrient absorption” is entered in all of the ten cell lines. To compare the ten cell lines this is meaningless, rather ruins the simplicity of the Table. How about deleting this phrase and adding a statement about “the usefulness of all the ten cell lines in studying drug and nutrient absorption” in the text (to the last paragraph - final conclusion in lines 466-)?

Thank you for pointing this out. We deleted the phrase “drug and nutrient absorption” from Table 3 except for HROC43, HROC183 T0 M2 and HROC217 T1 M2 and added a statement regarding the potential application of the best suited HROC model candidates for nutrient absorption studies to the conclusion part. Thanks for this suggestion. It might indeed attract more readers.

Reviewer 2 Report

The manuscript by Przybylla and Coll. focuses on the characterization of ten immortalized cell lines derived from human colorectal tumors. While some features of these lines, like the epithelial barrier properties, were reported in a previous paper of the same research team, the present study describes a deepen characterization including their differentiation in the main subtypes of intestinal epithelial cells, the expression of a panel of intestinal epithelial genes, by RT-qPCR, IHC or proteomics, their regulation by Rifampicin and Vitamin D3, and the imaging of the mature epithelium by SEM microscopy. All of these properties were compared with those of the well-known CaCo-2 cell line, the traditional standard for in vitro modeling the intestinal epithelial functions.

Although the manuscript is well-written and provides useful information for this research field, the abundance of images makes it less readable. We would suggest leaving only a set of the most representative pictures, providing all the details in the supplementary information. Also, be sure that tables #1 and #3 are properly formatted for publications.

The biological meaning of the correlation analysis reported in Figure 8 is confusing. Please include an explanation of the results and a comment in the discussion or remove it.

Minor corrections

Raw 60: Rifampin (Rifampicin?)

Raw 82: For (delete)

Raw 83: quantative (quantitative)

Raws 182-184 “Cells were grown … Hoechst staining” (remove redundant information already reported in M&M)

The same for figures #3 (Raws 213-215), #4 (Raws 234-236), #5 (Raws260-262)

Raw 226 und (and)

Raw 291 my (may)

Author Response

Answers to Reviewer 2

The manuscript by Przybylla and Coll. focuses on the characterization of ten immortalized cell lines derived from human colorectal tumors. While some features of these lines, like the epithelial barrier properties, were reported in a previous paper of the same research team, the present study describes a deepen characterization including their differentiation in the main subtypes of intestinal epithelial cells, the expression of a panel of intestinal epithelial genes, by RT-qPCR, IHC or proteomics, their regulation by Rifampicin and Vitamin D3, and the imaging of the mature epithelium by SEM microscopy. All of these properties were compared with those of the well-known CaCo-2 cell line, the traditional standard for in vitro modeling the intestinal epithelial functions.

Although the manuscript is well-written and provides useful information for this research field, the abundance of images makes it less readable. We would suggest leaving only a set of the most representative pictures, providing all the details in the supplementary information. Also, be sure that tables #1 and #3 are properly formatted for publications.

Thanks for your feedback. Table 1 was adjusted according to your advice. We further decided to move immunofluorescence images of MUC2, LYZ and CHGA staining into the supplement.

The biological meaning of the correlation analysis reported in Figure 8 is confusing. Please include an explanation of the results and a comment in the discussion or remove it.

 Explanations for the meaning of this correlation analysis was added as follows:

 “Key aspect of novel human intestinal in vitro models’ characterization was the detection of relevant transporter and nuclear receptor expression. Correlation analysis was done to examine the simultaneous expression of relevant DTs and NRs to deliver evidence for functional gene-gene interaction networks active in these lines.“

Minor corrections

Raw 60: Rifampin (Rifampicin?)

Raw 82: For (delete)

Raw 83: quantative (quantitative)

Raws 182-184 “Cells were grown … Hoechst staining” (remove redundant information already reported in M&M)

The same for figures #3 (Raws 213-215), #4 (Raws 234-236), #5 (Raws260-262)

Raw 226 und (and)

Raw 291 my (may)

These changes were made in order to your advice. We apologize for these minor errors.

Reviewer 3 Report

Manuscript cells-2607006 peer-review v1

Title: Novel in Vitro Models for Cell Differentiation and Drug

Transport Studies of the Human Intestine

Brief summary

Authors characterized ten low-passaged patient derived colorectal cancer cell lines for being used in in vitro assays of absorption, distribution, metabolism and excretion (ADME) and these lines were compared with the Caco-2 cell line derived from human colorectal adenocarcinoma used widely in most in vitro models for several intestinal studies. Cell lines assessment included markers of cell differentiation, RNA and protein expression of drug transporters and morphologic and functional features were also characterized. Data evidenced that compared to the Caco-2 cells, three human cell lines displayed both higher resemblance to jejunal epithelial tissue and higher regulatory potential of prominent drug transporters. Findings suggest potential use of the colonic human cell lines for ADME that share higher similarity regarding gene and protein expression of jejunal tissue than Caco-2 cells.

Highlights

This is a very concise, informative, readily for reading and very well written article that provides data critically discussed and importantly limitations were also included. Findings may support the use of the human colorectal cancer cell lines as novel in vitro models for basic and preclinical studies including ADME assays frequently tested in Caco-2 cell cultures.

Minor comments:

1. clarify please why human cell lines were incubated for 21 days as described previously (Przybylla et al 2022 DOI: 10.3390/ijms23179861) and in the current manuscript cell lines were incubated for 10 days post confluency (line294)

2. clarify please whether the findings raised from colonic cell lines might mirror the conditions of jejunal environment where most extent of nutrient transport takes place

Author Response

Answers to Reviewer 3

Brief summary

Authors characterized ten low-passaged patient derived colorectal cancer cell lines for being used in in vitro assays of absorption, distribution, metabolism and excretion (ADME) and these lines were compared with the Caco-2 cell line derived from human colorectal adenocarcinoma used widely in most in vitro models for several intestinal studies. Cell lines assessment included markers of cell differentiation, RNA and protein expression of drug transporters and morphologic and functional features were also characterized. Data evidenced that compared to the Caco-2 cells, three human cell lines displayed both higher resemblance to jejunal epithelial tissue and higher regulatory potential of prominent drug transporters. Findings suggest potential use of the colonic human cell lines for ADME that share higher similarity regarding gene and protein expression of jejunal tissue than Caco-2 cells.

Highlights

This is a very concise, informative, readily for reading and very well written article that provides data critically discussed and importantly limitations were also included. Findings may support the use of the human colorectal cancer cell lines as novel in vitro models for basic and preclinical studies including ADME assays frequently tested in Caco-2 cell cultures.

Minor comments:

  1. clarify please why human cell lines were incubated for 21 days as described previously (Przybylla et al 2022 DOI: 10.3390/ijms23179861 ) and in the current manuscript cell lines were incubated for 10 days post confluency (line294)

Thanks for your comment. The discussion part was supplemented as follows:

“In our previous study, HROC cell lines were incubated for 21 days to perform TEER measurements and FITC dextran permeability tests according to the protocols used for Caco-2 (https://doi.org/10.3762/bjnano.5.239, https://doi.org/10.1016/j.bbrep.2022.101314). Our current study aimed to standardize a protocol for the HROC lines with shorter culture time periods necessary for differentiation into IECs and application as intestinal epithelial models for transport studies. Thus, these lines were grown for ten days post-confluence instead of 21 days necessary for Caco-2 (https://doi.org/10.1038/nprot.2007.303.nprot.2007.303).”

  1. clarify please whether the findings raised from colonic cell lines might mirror the conditions of jejunal environment where most extent of nutrient transport takes place

We again thank you for your feedback. The discussion was supplemented as follows:

“Despite their colonic origin, HROC lines might differ in mirroring jejunal properties due to different primary tumor localization sides. This might explain jejunal properties remaining in a subset of cell lines and eventually resulted in a higher variance of small intestinal properties among these lines.“

Round 2

Reviewer 1 Report

The manuscript has been properly revised. I appreciate the authors' efforts. This is a quite informative paper and will be useful for the readers working in this research field.